# Effects of Configuration Mode on the Light-Response Characteristics and Dry Matter Accumulation of Cotton under Jujube–Cotton Intercropping

**Tiantian Li [1], Peijuan Wang [1], Yanfang Li [1], Ling Li [1], Ruiya Kong [1], Wenxia Fan [1], Wen Yin [2], Zhilong Fan [2], Quanzhong Wu [1], Yunlong Zhai [1], Guodong Chen [1,*] and Sumei Wan [1,*]**

[1]   College of Agriculture, Tarim University, Alar 843300, China
[2]   College of Agronomy, Gansu Agricultural University, Lanzhou 730070, China
*   Correspondence: guodongchen@taru.edu.cn (G.C.); wansumei@taru.edu.cn (S.W.)

**Abstract:** The current study evaluated the canopy cover competition for light and heat in a jujube–cotton intercropping system to measure the growth and yield performance of cotton, and the optimal cotton planting configuration. In this study, a two-year field experiment (2020 and 2021) was studied with different spacing configuration modes designed as follows: two rows of cotton (CM1) planted 1.4 m apart, four rows of cotton (CM2) planted 1.0 m apart, and six rows of cotton (CM3) planted 0.5 m apart, spacing intercropped jujube trees, respectively. The control treatment consisted of monocultured cotton (CK). The light-response curve was plotted using an LI-6400 XT photosynthesis instrument. Based on the modified rectangular hyperbola model, the photosynthetic characteristics were fitted, and the dry matter distribution characteristics and yield were compared. The results showed that with the increase in photosynthetically active radiation, the net photosynthetic rate (Pn) of each growth phase decreased first and then increased rapidly in the range of 0–200 $\mu mol \cdot m^{-2} \cdot s^{-1}$ and then decreased slightly after the inflection point (light saturation point). The light-response curves of stomatal conductance and transpiration rate showed a linear relationship. The trend in the intercellular $CO_2$ concentration response curve was opposite to that of Pn. The maximum Pn (Pmax) of intercropped cotton was significantly impacted by configuration modes, of which CM2 treatment generated 1.8% and 22.8% higher Pmax than the CM1 and CM3 treatments. The cotton yield in the two years ranked as CK > CM3 > CM2 > CM1, and the average land equivalent ratio of CM2 was significantly higher than that of CM3 (22.4%) and CM1 (95.9%). The six-row configuration resulted in greater competition with the trees, which affected the accumulation of below-ground dry matter, while the four-row configuration formed a reasonable canopy structure, which ensured that more photosynthetic substances were distributed to the generative organs. The reasonable four-rows configuration mode may improve the photosynthetic efficiency of intercropped cotton economic yield.

**Keywords:** dry matter; jujube–cotton intercropping; light-response curve; production; rectangular hyperbolic correction model

## 1. Introduction

Xinjiang (34°22′–49°50′ N, 73°40′–96°23′ E, altitude 115–8611 m) is located in the northwest of China. Although the climate is dry and the environment is harsh, its vast territory and abundant light and heat resources play a decisive role in China's cotton and fruit industry. Xinjiang is the largest high-yield and high-quality cotton production base in China [1,2]. In 2021, the Xinjiang cotton planting area was 2506.1 thousand hectares and the yield per unit area was 2.1 t/ha; cotton production reached 512.9 million tons, accounting for 89.5% of the total national cotton production. Intercropping of fruiting trees with cultivated areas of grain or cotton may explore and contribute to food security and

sustainability. Agroforestry systems that combine annual crops with trees have been widely used in semi-arid regions [3], and intercropping is currently implemented in Africa, Asia, Europe, and the Americas [4]. Intercropping allows crops to find their niche in time and space and promotes interspecific complementation to increase economic yield [5]. However, when crops require the same resources at the same time and space, then competition occurs. Growing cotton between rows of jujube trees as a companion crop is a common and profitable practice, especially during the young growth phase of jujube trees (less than 10 years). The jujube–cotton intercropping system is a complex system with high environmental heterogeneity [6]. The most intense competition factor between cotton crops and jujube trees in the intercropping system is light [7]. The blocking of light by high crops to low crops will affect the photosynthetic distribution above the canopy of the low crops, which will alter the photosynthetic characteristics and reduce the economic yield [8]. Approximately 90–95% of crop dry matter accumulation originates from photosynthesis. Existing studies have found that the main limiting factors of intercropping systems are field management, planting structure, density, and environmental conditions [9–12]. The planting structure of jujube–cotton intercropping determines the resource competition, which affects the potential resource capture and optimal allocation of the system and directly affects the crop canopy structure, thereby influencing its photosynthesis and yield formation [13,14]. Therefore, minimizing the resource competition between jujube and cotton and optimizing the allocation of light and heat resources are key to improving the yield and overall productivity of jujube–cotton intercropping systems.

In the process of photosynthesis, the quantitative relationship between net photosynthetic rate (Pn) and photosynthetically active radiation (PAR) is the basis for revealing the instantaneous response of plant photosynthetic physiological processes to the environment [15–17], which is called the light response. It is the most effective method for studying the influence of the environment on plant photosynthesis. It is very important to understand the photo-physiological basis of the photosynthesis process and predict the change trend in photosynthetic rate under light conditions in order to explain crop productivity [18,19]. The light-response curve model is the most effective means for assessing the light response of plants affected by the environment. The corresponding physiological parameters can be obtained by analyzing the fitted light-response curve. To date, researchers have established a variety of light-response models. Common light-response models include the rectangular hyperbolic model [20], non-rectangular hyperbolic model [21], exponential model [22], and rectangular hyperbolic correction model [23]. Previous studies [24,25] found that the rectangular hyperbolic correction model can better estimate the saturated light intensity than other models. The simulation of this model is a function with extreme value and the simulation degree is high, and its fitting effect is closer to the actual value than other models [26].

It is still not clear to what extent the growth and development of cotton can be improved by adjusting the row spacing configuration to reduce interspecific competition in jujube–cotton intercropping systems in Northwest China. Therefore, in this study, field plot experiments with different configurations were carried out in a jujube–cotton intercropping system in southern Xinjiang to study the effects of different configuration modes on the light-response curves of cotton. The objectives of this jujube–cotton intercropping system study were (I) to identify the best fitted light-response parameters, (II) to clarify light-response curve change trend, and (III) to analyze its photosynthetic physiological characteristics, (IV) to find the best configuration mode for the dry matter accumulation of cotton.

## 2. Materials and Methods

### 2.1. Test Area and Test Material Overview

The experiment was conducted from April 2020 to October 2021 at the Tarim Horticultural Experiment Station, affiliated with Tarim University (40°32′34″ N, 81°18′07″ E, altitude 1015 m) (Figure 1). The study area is located in the upper reaches of the Tarim River and the northwest edge of the Taklimakan Desert. It belongs to the arid desert climate of

the warm temperate continent and is rich in light and heat resources. The average annual solar radiation is 559.4–612.1 KJ·cm$^{-2}$, the average annual sunshine is 2556.3–2991.8 h, and the sunshine rate is 66%. The rainfall and average temperature during the growing season in 2020 and 2021 are shown in Figure 2. The soil type is sandy loam, soil pH 7.9, drip irrigation. The test jujube (*Ziziphus jujuba* L.) (sour jujube) orchard was established by direct seeding in 2012, and jujube (gray jujube) was grafted in the spring of 2014 [27]. The plant spacing of jujube was 3 m × 1 m (row spacing × plant spacing). The tested cotton (*G. hirsutum* L.) variety was 'Xinluzhong 82' (the seeds sown on 10 April 2020).

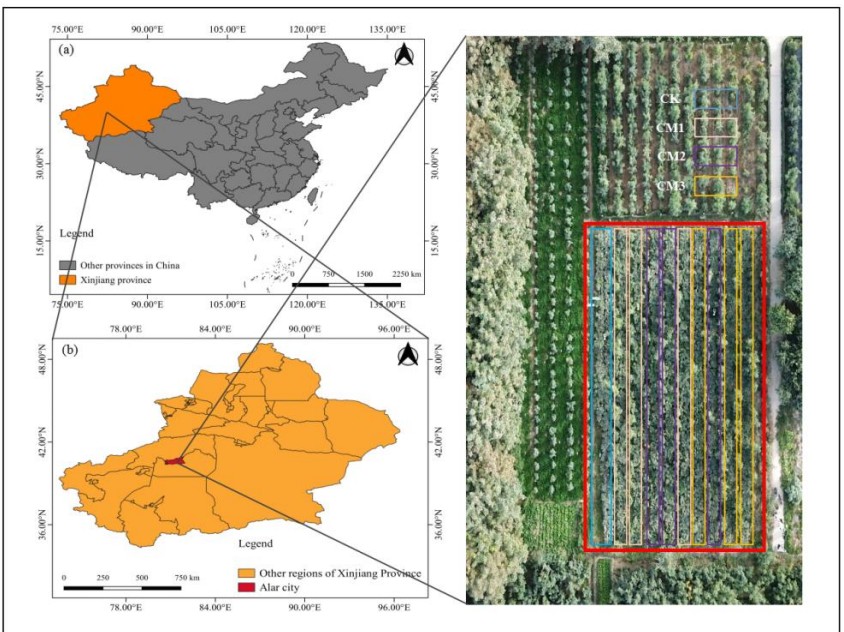

**Figure 1.** Study area map. (**a**) The location of Xinjiang province; (**b**) the location of the Alar city site; (**c**) the image of the experimental field (red box area) obtained on 16 September 2021 using a unmanned air vehicle.

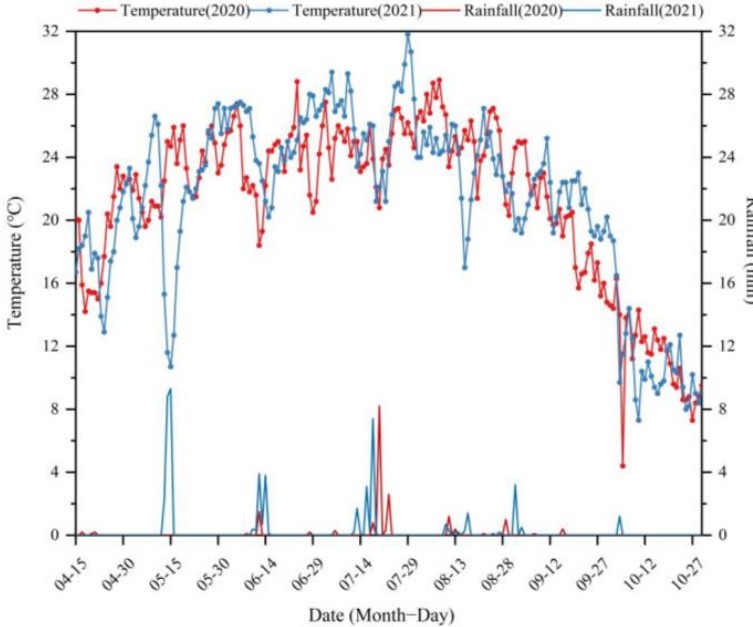

**Figure 2.** The rainfall and daily average temperature of the experimental field in 2020 and 2021.

### 2.2. Experimental Design

In the experiment, the number of rows and the spacing of the cotton plants planted between rows of jujube trees (the distance between the cotton line adjacent to the jujube tree and the jujube tree trunk) were configured as the research factor. Three different field configuration modes (Figure 3) were set up. Two rows of cotton (CM1) were planted at a distance of 1.4 m from the jujube trees, and the row spacing was 20 cm. Four rows of cotton (CM2) were planted at a distance of 1.0 m from the jujube trees, and the row spacing was (20 + 60 + 20) cm. Six rows of cotton (CM3) were planted 0.5 m away from the jujube trees, and the row spacing was (20 + 70 + 20 + 70 + 20) cm. At the same time, monocultured cotton (CK) was set as the control treatment with a row spacing of (66 + 10) cm. The experiment consisted of four treatments, and each treatment had three replicates. There was a total of 12 plots with a plot area of 120 m², and the plots were randomly arranged.

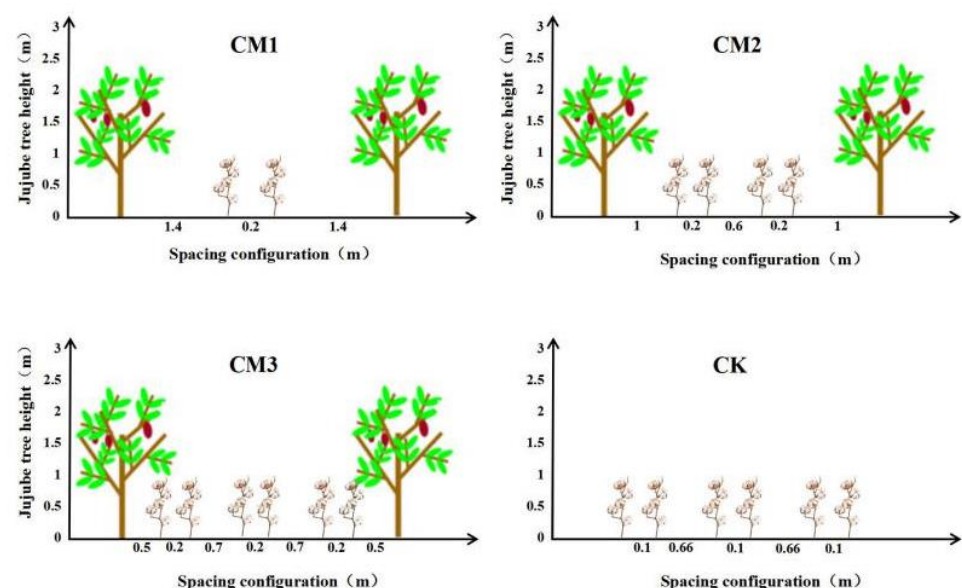

**Figure 3.** Diagram of field configuration modes.

### 2.3. Method

2.3.1. Plotting of Light-Response Curve

The light-response curves were plotted by using an LI-6400XT portable photosynthesis system (Licor Inc., Lincoln, NE, USA). Measurements were taken during the 2021 cotton seedling period (18 d after sowing), budding period (55 d after sowing), flowering and boll development period (86 d after sowing), and maturation period (145 d after sowing) on windless sunny days between 11:00 and 13:00. Three plants with consistent growth status were selected. The fully unfolded third leaves were measured, and three biological replicates were measured for each configuration treatment. To reduce environmental interference, the open-air path was used, and the parameters were set as follows: air flow rate 500 $\mu mol \cdot mol^{-1}$, $CO_2$ concentration $(400 \pm 20)$ $\mu mol \cdot mol^{-1}$, leaf temperature $(25 \pm 2)$ °C, and relative humidity $(50 \pm 5)$%. The gradient of PAR simulated by red and blue light sources from weak to strong was 0, 200, 400, 600, 800, 1000, 1200, 1400, 1600, 1800, and 2000 $\mu mol \cdot m^{-2} \cdot s^{-1}$, with a total of 11 gradients. Before each measurement, the leaves were induced for approximately 60–120 s under 1200 $\mu mol \cdot m^{-2} \cdot s^{-1}$ light intensity, and the minimum stable time after light intensity change was set to 180 s. When the variation rate of the measurement results was less than 0.05, measurements were automatically recorded by the instrument [28,29].

The measurement items included the net photosynthetic rate (Pn, $\mu mol \cdot m^{-2} \cdot s^{-1}$), transpiration rate (Tr, $mmol \cdot m^{-2} \cdot s^{-1}$), stomatal conductance (Gs, $mol \cdot m^{-2} \cdot s^{-1}$), intercellular $CO_2$ concentration (Ci, $\mu mol \cdot mol^{-1}$), and leaf instantaneous water-use efficiency

(WUE, µmol·mmol$^{-1}$) = Pn/Tr, and the light-response curve was fitted using the photosynthetic calculation software (4.1.1) (CHN Jiangxi, jinggangshan university, college of life sciences, Ye, Z.) [30]. The best-fitting effect of the rectangular hyperbolic correction model was selected [24]. The fitting effect of the rectangular hyperbolic correction model is shown in Figure 4, and the R$^2$ value is 0.9996. The main physiological parameters, including maximum Pn (Pmax), apparent quantum efficiency (AQE), light saturation point (LSP), light compensation point (LCP), dark respiration rate (Rd), and determination coefficient (R$^2$), were also obtained. The fitting model is given by

$$Pn = \alpha \frac{1 - \beta I}{1 + \gamma I} I - Rd \tag{1}$$

where $\alpha$ is the initial slope of the light-response curve, $\beta$ and $\gamma$ are coefficients, I is the PAR, and Rd is dark respiration.

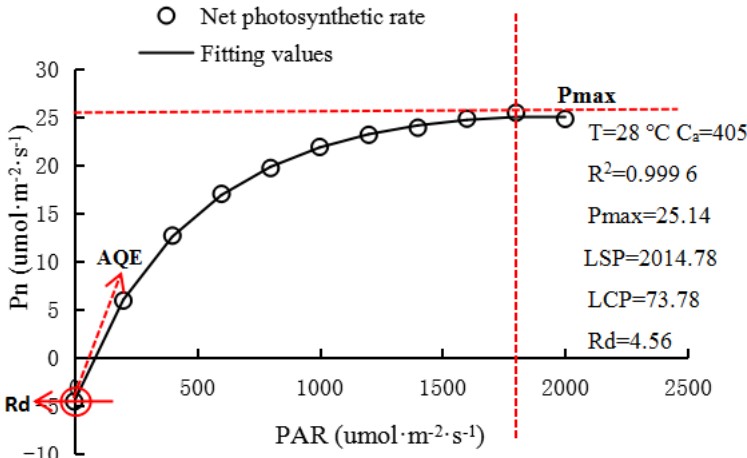

**Figure 4.** The rectangular hyperbolic correction model fits the light-response curve. (T: leaf temperature; Ca: reference chamber $CO_2$ concentration; R$^2$: coefficient of determination; Pmax: the maximum net photosynthetic rate; LSP: light saturation point; LCP: light compensation point; Rd: dark respiration rate; AQE: apparent quantum efficiency).

LSP is given by

$$LSP = \frac{\sqrt{(\beta + \gamma)/\beta} - 1}{\gamma} \tag{2}$$

Pmax is given by

$$Pmax = \alpha \left( \frac{\sqrt{\beta + \gamma} - \sqrt{\beta}}{\gamma} \right)^2 - Rd \tag{3}$$

### 2.3.2. Dry Matter Determination

Destructive sampling was carried out in each replicate of the different configuration modes in each cotton growth period, and three representative cotton plants were selected. A total of nine plants were selected in each configuration to minimize errors and collect data on the dry matter accumulation of cotton at different growth periods. Different organs (root, stem, leaf, bud, flower, boll) were sampled and dried at 105 °C in an oven for 30 min and then dried at 80 °C until a constant mass to determine the dry matter yield [31].

### 2.3.3. Determination of Cotton Economic Yield and Its Components

Final cotton ball harvesting was carried out on October 16, 2020, and on October 18, 2021, respectively. The numbers of plants and bolls were determined in a specific area in the test area, and 50 bolls were taken from the upper (average 85 cm above ground), middle (average 53 cm above ground), and lower (average 20 cm above ground) parts of the cotton

plants in each plot to measure the boll weight and lint percentage. Three replicates were measured in each plot. After separating the cotton wool and cottonseeds, the seed cotton and lint cotton of each treatment were weighed to calculate the lint percentage, and the fruits of intercropped jujube were harvested at the same time.

The land equivalent ratio (LER) is an index to evaluate the environmental resource utilization efficiency and yield change of intercropping compared with monocropping, and it was calculated as follows [32]:

$$LER = \frac{Y_{ic}}{Y_{sc}} + \frac{Y_{ij}}{Y_{sj}} \tag{4}$$

where $Y_{ic}$ and $Y_{sc}$ are the yield of intercropping and monocropping cotton, respectively, and $Y_{ij}$ and $Y_{sj}$ are the yield of intercropping and monocropping jujube, respectively. When LER > 1, it shows that intercropping has a yield advantage.

### 2.4. Statistical Analysis

MS Excel v. 2021 was used to sort out the original data and to draw tables. With the help of SPSS 26.0 (SPSS Inc., Chicago, IL, USA), the single factor least-significant difference method (LSD) was used to test the statistical significance of the data at the $p < 0.05$ level, and Origin 2021 (Origin Lab Corporation, Northampton, MA, USA) was used to plot the light-response curve, and we used Quantum GIS 3.8.3 (Open Source eospatial Foundation) to draw the test location map.

## 3. Results
### 3.1. Differences in the Cotton Light-Response Curves under Different Configuration Modes
3.1.1. Cotton Pn Light-Response Curves

Under different configuration modes, the response of cotton Pn to PAR showed a trend of an initial rapid increase and then a slow increase, following which it stabilized. It then decreased slightly after the inflection point (light saturation point), and the trend in the light response at each growth period was similar (Figure 5). In the range of 0–200 $\mu mol \cdot m^{-2} \cdot s^{-1}$ (i.e., the range above the light compensation point), Pn increased proportionally and rapidly, indicating that PAR was the main limiting factor for photosynthesis in this range. When the light intensity exceeded a certain value, the growth trend in Pn began to decrease. Due to the light saturation point being reached, the cotton leaves could not absorb and utilize high light intensity, which limited the photosynthetic rate. During the seedling period, Pn clearly increased linearly with light intensity. At Pmax, there were significant differences among the treatments. Due to the absence of shading effect in monocropping, the Pmax value of the CK (28.73) treatment was significantly higher than that of the other treatments, while the CM3 (16.45) treatment reached Pmax at the light saturation point. The photosynthetic rate of each treatment was the highest at the flowering and boll development period. Under the same PAR throughout the growing season, the Pn of cotton was the largest with the CK treatment and the smallest with the CM3 treatment, indicating that the shade provided by the jujube trees during intercropping affected the net photosynthetic rate of cotton. The closer the distance to jujube trees was, the greater the shading effect became.

3.1.2. Cotton Tr Light-Response Curves

The response curves of transpiration rate to PAR in different growth periods are shown in Figure 6. There was a linear correlation between transpiration rate and PAR. The Tr of each treatment throughout the entire growth period was as follows: seedling period > flowering and boll development period > budding period > maturation period. The lowest Tr in the maturation period was mainly due to the decline and shedding of the cotton leaves with the growth period. Among the four growth periods, CM3 showed the lowest Tr, with the largest difference observed during the seedling period and the smallest difference observed during the maturation period compared with the other treatments. The Tr of the

CM2 treatment was significantly higher than that of the CM3 treatment during the budding period and flowering and boll development period.

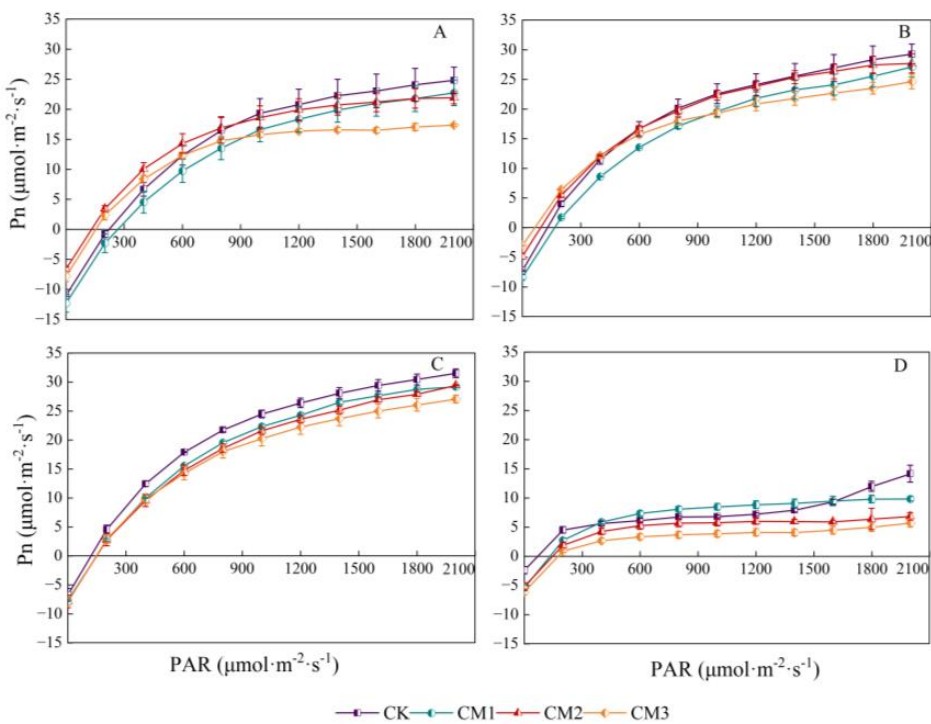

**Figure 5.** Light-response curves of cotton net photosynthetic rate (Pn) under different configuration modes. (**A**: seedling period; **B**: budding period; **C**: flowering and boll development period; **D**: maturation period.).

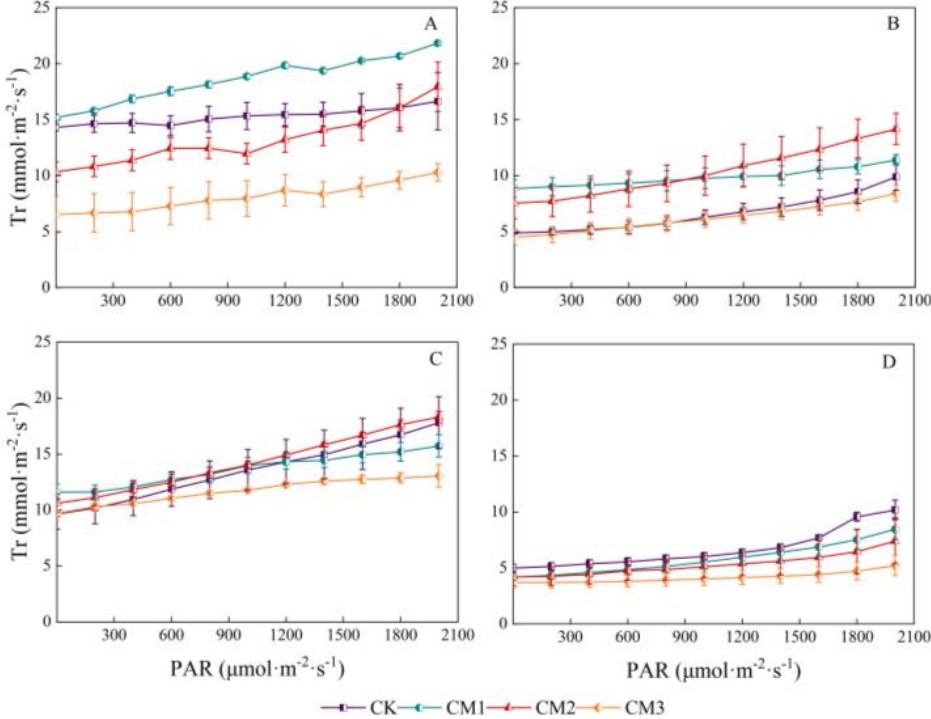

**Figure 6.** Light-response curves of cotton transpiration rate (Tr) under different configuration modes. (**A**): seedling period; (**B**): budding period; (**C**): flowering and boll development period; (**D**): maturation period.).

### 3.1.3. Cotton Gs Light-Response Curves

There was a significant linear relationship between stomatal conductance and PAR (Figure 7). The stomatal conductance of each treatment increased linearly with the increase in PPDD in the range of 0–2000 $\mu mol \cdot m^{-2} \cdot s^{-1}$, which was consistent with the trend in the Tr light-response curve. The response curves of the different treatments varied with different growth periods. Gs at the seedling period showed the following order: CM1 > CK > CM2 > CM3; the Gs of the CM2 treatment at the budding period was the largest; with no shading in monocropping system, Gs of CK treatment was significantly higher than that of the other three treatments; and there was no significant difference between CM1, CM2, and CM3. The curves of CM2 and CM3 were consistent during the maturation period.

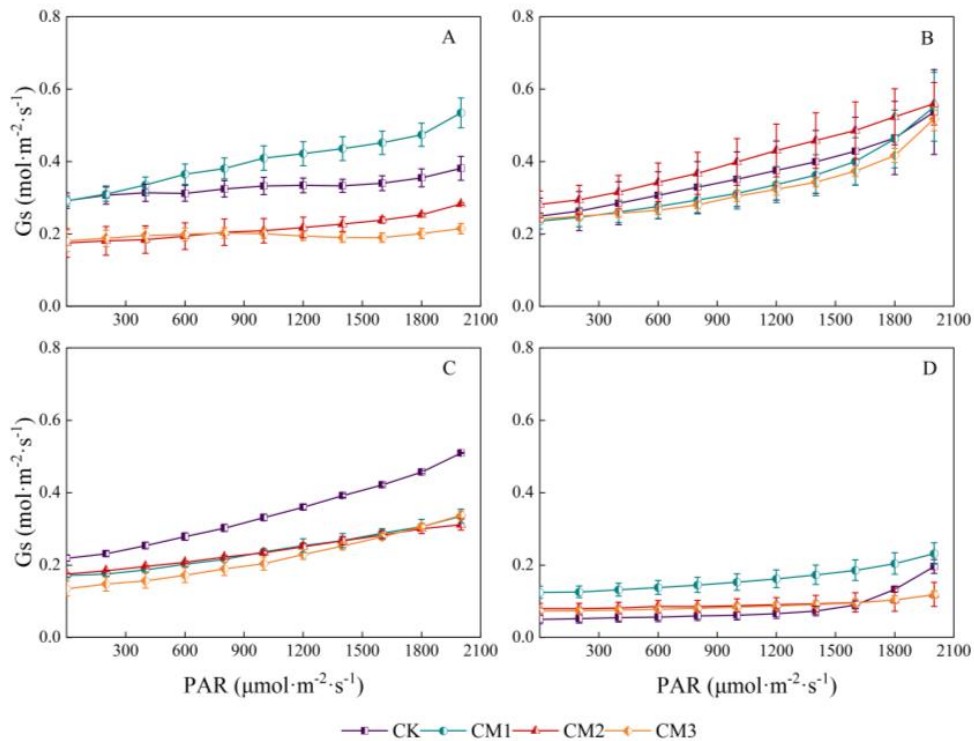

**Figure 7.** Light-response curves of cotton stomatal conductance (Gs) under different configuration modes. (**A**): seedling period; (**B**): budding period; (**C**): flowering and boll development period; (**D**): maturation period.)

### 3.1.4. Cotton Ci Light-Response Curves

Contrary to Pn, at an intercellular $CO_2$ concentration in the range of 0–200 $\mu mol \cdot m^{-2} \cdot s^{-1}$, the light-response curve decreased rapidly with the increase in light intensity, following which it stabilized and then increased slowly (Figure 8). The rapid decrease in Ci indicates that the photosynthetic process had a large demand for photosynthetic raw material $CO_2$ and high photosynthetic efficiency. As the increase in Gs was small under low PAR, and the increase in Pn rapidly increased the consumption of $CO_2$, the $CO_2$ lagged behind with the increase in the photosynthetic rate, thereby resulting in stomatal limitation. With the continuous increase in light intensity, the stomata were stimulated and opened, the concentration of $CO_2$ entering from the air increased, and Ci increased. Except for the seedling period, the $CO_2$ concentration of CK and CM2 in the other growth periods was the lowest, indicating that the CK and CM2 treatments had the highest utilization rate of light energy in this experiment. With the continuation of growth, the concentration of Ci in the cotton leaves increased gradually, which may be caused by the decrease in the photosynthetic rate of the senescent leaves.

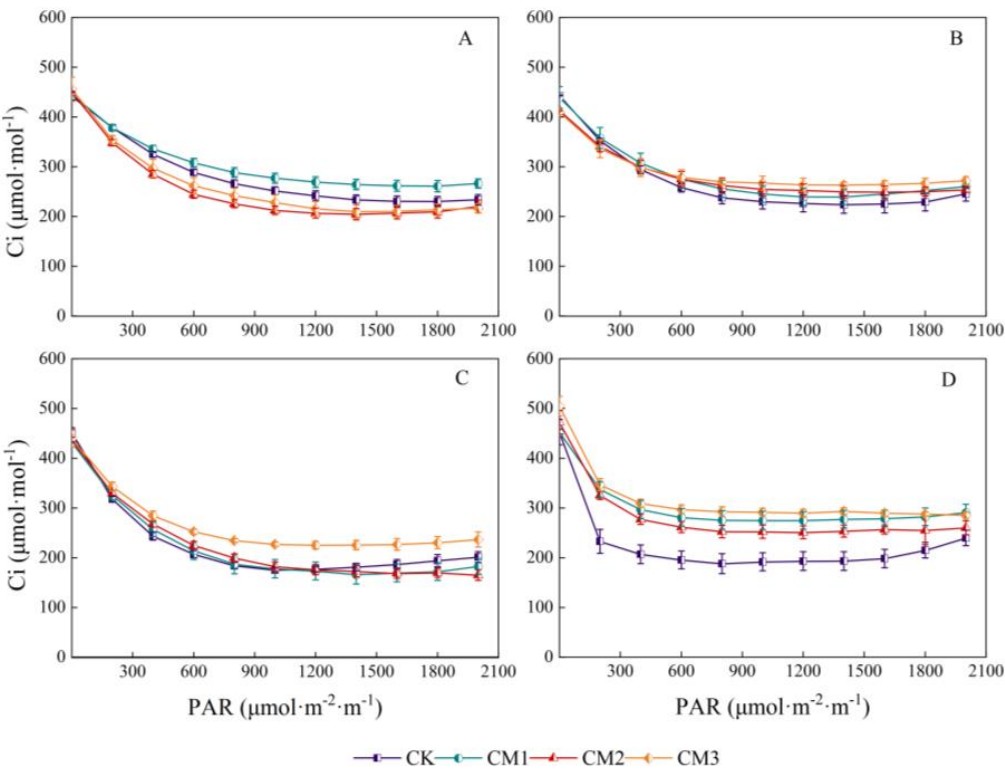

**Figure 8.** Light-response curves of cotton intercellular $CO_2$ concentration (Ci) under different configuration modes. (**A**): seedling period; (**B**): budding period; (**C**): flowering and boll development period; (**D**): maturation period.).

### 3.1.5. Cotton WUE Light-Response Curves

There was a significant parabolic correlation between leaf instantaneous WUE and PAR. With the rapid increase in PAR, leaf instantaneous WUE tended to stabilize and then began to decrease (Figure 9). It increased rapidly in the range of low light intensity $0$–$200$ $\mu mol \cdot m^{-2} \cdot s^{-1}$ and began to decrease at $1500$ $\mu mol \cdot m^{-2} \cdot s^{-1}$. The reason may be that excessive PAR stimulated the opening of the stomata and the strengthening of the transpiration rate, while the ability of photosynthetic C assimilation did not increase proportionately, resulting in a decrease in WUE. Each configuration treatment showed significant differences in the budding period and the flowering and boll development period, and there were no significant differences in the maturation period of the senescent cotton leaves. The WUE of CK and CM1 was higher in each growth period; followed by CM2, the WUE of CM3 was low. The trend in the WUE response curve at the budding period and the flowering and boll development period was similar, obtaining the highest values; followed by the seedling period, WUE was the lowest at the maturation period.

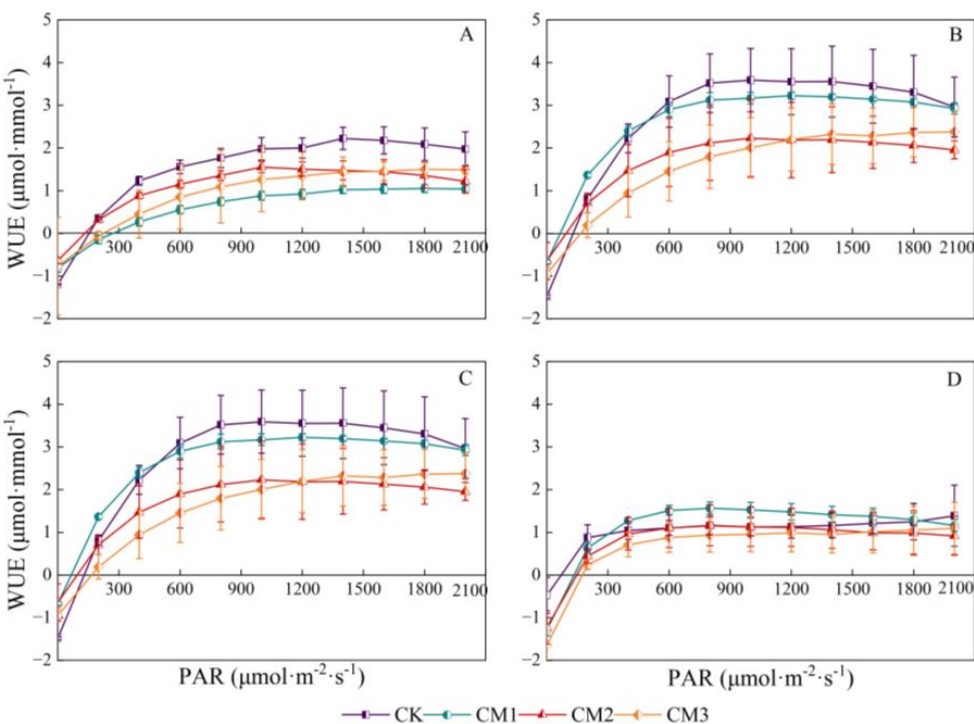

**Figure 9.** Light-response curves of cotton leaf instantaneous water-use efficiency (WUE, μmol·mmol$^{-1}$) under different configuration modes. (**A**): seedling period; (**B**): budding period; (**C**): flowering and boll development period; (**D**): maturation period.).

### 3.2. Effects of Different Configuration Modes on Cotton Photosynthetic Characteristic Parameters

Using the photosynthetic calculation assistant software, the light-response fitting parameters of the different treatments at each growth period were obtained by using the rectangular hyperbolic correction model (Table 1). The light-response model of each configuration treatment was well fitted and reached a very significant level, but the effects of the fitted light-response curve parameters were different. Among them, the LSP of the CK treatment during the cotton seedling period and the budding period was the highest at 1996.6 μmol·m$^{-2}$·s$^{-1}$ and 3038.2 μmol·m$^{-2}$·s$^{-1}$, respectively. The LSP of the CM2 treatment was the largest during the flowering and boll development periods, reaching 2618.3 μmol·m$^{-2}$·s$^{-1}$ and 1416.1 μmol·m$^{-2}$·s$^{-1}$, respectively, indicating that the corresponding light intensity utilization was large. When PAR increased, the CM2 treatment could make full use of the light conditions. On the contrary, the LSP of the CM3 treatment was the lowest throughout the growing season. With the progression of growth, the LSP decreased, indicating that the ability of the cotton leaves to use external light decreased over time. LCP reflects the light intensity when the photosynthetic assimilation of the cotton leaves is equivalent to the respiration consumption. With the extension of the growth period, LCP in the CK and CM1 treatments gradually decreased first and then increased in the CM2 treatment. This indicates that the utilization rate of low light by the CM2 treatment increased in the late period of cotton growth. The maximum photosynthetic value Pmax depends on the Rubisco activity and electron transport rate [33]. The CK treatment had a higher Pmax throughout the whole growth period, followed by CM2 (except for the maturation period). This suggests that the Rubisco activity of CK and CM2 was high, and the electron transfer rate was fast, which is beneficial to dry matter accumulation in cotton. At a lower PAR (below 200 μmol·m$^{-2}$·s$^{-1}$), there was no significant difference in the performance of AQE during the first three growth periods. The ability of each treatment to utilize low light intensity was similar, but significant differences began to appear during the maturation period. The ability of the CM2 and CM1 treatments to utilize low light was significantly higher than that of the other treatments. Rd represents the dark respiration

rate of cotton leaves. The CK and CM1 treatments decreased Rd with the senescence of the cotton leaves. The trend in Rd in the CM2 treatment did not decrease but rather increased, which may be due to the increase in AQE. The main site of Rd is in the mitochondria and cytoplasm, and the mitochondria are also one of the sites for ATP synthesis. This is one of the reasons why CM2 still maintained a high photosynthetic rate in the later period.

**Table 1.** Cotton light-response characteristic parameters at different growth periods.

| Period of Growth | Treatments | LSP ($\mu mol \cdot m^{-2} \cdot m^{-1}$) | LCP ($\mu mol \cdot m^{-2} \cdot m^{-1}$) | Pmax ($\mu mol \cdot m^{-2} \cdot m^{-1}$) | AQE ($\mu mol \cdot mol^{-1}$) | Rd ($\mu mol \cdot m^{-2} \cdot m^{-1}$) | $R^2$ |
|---|---|---|---|---|---|---|---|
| Seedling period | CK | 1996.57 ± 165.05 [a] | 240.03 ± 38.52 [a] | 28.73 ± 6.00 [a] | 0.062 ± 0.005 [a] | 11.327 ± 0.194 [a] | 0.9982 ** |
| | CM1 | 1801.75 ± 704.29 [a] | 233.20 ± 64.34 [a] | 19.51 ± 3.32 [bc] | 0.060 ± 0.002 [a] | 11.079 ± 2.556 [a] | 0.9997 ** |
| | CM2 | 1947.74 ± 400.54 [a] | 162.79 ± 48.77 [a] | 25.09 ± 3.60 [ab] | 0.060 ± 0.019 [a] | 6.708 ± 1.058 [b] | 0.9917 ** |
| | CM3 | 1683.31 ± 27.07 [a] | 173.15 ± 32.19 [a] | 16.45 ± 0.65 [c] | 0.073 ± 0.002 [a] | 9.662 ± 1.561 [ab] | 0.9973 ** |
| Budding period | CK | 3038.15 ± 2060.43 [a] | 126.76 ± 16.04 [a] | 32.93 ± 2.30 [a] | 0.075 ± 0.004 [a] | 7.184 ± 0.195 [b] | 0.9977 ** |
| | CM1 | 2507.27 ± 843.79 [a] | 152.50 ± 23.87 [a] | 26.83 ± 5.16 [ab] | 0.071 ± 0.005 [a] | 8.247 ± 0.647 [a] | 0.9971 ** |
| | CM2 | 2949.83 ± 354.06 [a] | 88.85 ± 14.14 [b] | 28.99 ± 3.29 [ab] | 0.067 ± 0.001 [a] | 5.311 ± 0.842 [c] | 0.9996 ** |
| | CM3 | 1562.67 ± 172.53 [a] | 64.17 ± 4.25 [b] | 24.22 ± 2.52 [b] | 0.072 ± 0.012 [a] | 3.050 ± 0.199 [d] | 0.9972 ** |
| Flowering and boll development period | CK | 2519.90 ± 678.66 [a] | 109.02 ± 26.79 [a] | 33.55 ± 0.65 [a] | 0.070 ± 0.003 [a] | 6.551 ± 1.815 [a] | 0.9998 ** |
| | CM1 | 2510.46 ± 276.07 [a] | 149.75 ± 8.76 [a] | 32.63 ± 3.86 [a] | 0.065 ± 0.003 [a] | 8.105 ± 0.361 [a] | 0.9999 ** |
| | CM2 | 2618.25 ± 332.77 [a] | 106.56 ± 44.38 [a] | 32.69 ± 3.74 [a] | 0.064 ± 0.0003 [a] | 5.771 ± 1.901 [a] | 0.9998 ** |
| | CM3 | 2048.91 ± 67.26 [a] | 122.42 ± 29.78 [a] | 31.19 ± 3.93 [a] | 0.067 ± 0.004 [a] | 6.880 ± 1.136 [a] | 0.9997 ** |
| Maturation period | CK | 1306.67 ± 489.02 [a] | 125.33 ± 40.07 [a] | 13.24 ± 1.89 [a] | 0.031 ± 0.012 [b] | 2.826 ± 1.049 [b] | 0.9222 ** |
| | CM1 | 782.67 ± 126.64 [a] | 94.67 ± 22.03 [a] | 12.55 ± 2.04 [a] | 0.083 ± 0.001 [a] | 4.386 ± 0.924 [ab] | 0.9822 ** |
| | CM2 | 1416.11 ± 1101.87 [a] | 136.09 ± 32.61 [a] | 6.38 ± 2.33 [b] | 0.077 ± 0.006 [a] | 5.785 ± 0.532 [a] | 0.9296 ** |
| | CM3 | 405.33 ± 34.02 [a] | 126.67 ± 31.07 [a] | 4.01 ± 1.46 [b] | 0.044 ± 0.041 [ab] | 4.092 ± 1.038 [ab] | 0.975 ** |

Note: The value is the mean ± standard deviation. Different lowercase letters indicate that the difference between different treatments of the same index is significant ($p < 0.05$), and ** indicates that the difference reached a very significant level ($p < 0.01$). LSP: light saturation point; LCP: light compensation point; Pmax: the maximum net photosynthetic rate; AQE: apparent quantum yield; Rd: dark respiration rate; $R^2$: coefficient of determination.

### 3.3. Effects of Different Configuration Modes on the Dry Matter Distribution Characteristics of Cotton

The dynamic process of dry matter accumulation and distribution in the aboveground part of cotton was basically the same (Table 2), and the proportion of dry matter distribution under different configurations at different growth periods differed. From the seedling period to the budding period, the cotton plants grew slowly, dry matter accumulation was slow, and only vegetative growth occurred, and dry matter began to increase in the reproductive organs (buds, flowers, bolls) during the budding period. From the budding period to the flowering and boll development period, the field water and fertilizer supply was sufficient to allow a rapid increase in dry matter accumulation during this critical period of reproductive growth. During the maturation period, the dry matter accumulation was the highest, and the distribution of dry matter in the reproductive organs exceeded that in the vegetative organs. However, when the supply of water and fertilizer in the field stopped, the vegetative growth of the cotton stagnated, and the rate of dry matter accumulation decreased. With the progression of the growth period, the proportion of dry matter distribution in the vegetative organs decreased gradually in the two years from 96.3–97.4% (2020) and 90.4–93.0% (2021) in the budding period to 45.7–49.2% (2020) and 41.7–60.3% (2021) in the maturation period. The distribution proportion of the reproductive organs showed a gradual increasing trend from 2.64–3.81% (2020) and 7.80–16.2% (2021) in the budding period to 50.8–54.3% (2020) and 45.6–58.4% (2021) at the maturation period.

**Table 2.** Dry matter accumulation and distribution per plant at different growth periods of cotton.

| Growth-Period | Position | 2020 | | | | | 2021 | | | | |
|---|---|---|---|---|---|---|---|---|---|---|---|
| | | Vegetative Organ | | Generative Organ | | Weight per Plant (g) | Vegetative Organ | | Generative Organ | | Weight per Plant (g) |
| | Field Configuration | Allocation (g) | Scale (%) | Allocation (g) | Scale (%) | | Allocation (g) | Scale (%) | Allocation (g) | Scale (%) | |
| Seedling period | CM1 | 0.99 b | 100.00 | — | — | 0.99 | 1.90 b | 100.00 | — | — | 1.90 b |
| | CM2 | 0.87 b | 100.00 | — | — | 0.87 | 2.37 b | 100.00 | — | — | 2.37 b |
| | CM3 | 0.81 b | 100.00 | — | — | 0.81 | 1.79 b | 100.00 | — | — | 1.79 b |
| | CK | 1.65 a | 100.00 | — | — | 1.65 | 3.42 a | 100.00 | — | — | 3.42 a |
| Budding period | CM1 | 6.38 b | 97.36 | 0.16 b | 2.64 | 6.56 b | 19.11 a | 90.37 | 2.26 b | 9.63 | 21.23 ab |
| | CM2 | 8.46 a | 96.79 | 0.28 ab | 3.21 | 8.74 a | 16.14 a | 91.84 | 1.69 b | 9.62 | 17.58 b |
| | CM3 | 4.36 c | 96.88 | 0.14 b | 3.11 | 4.50 c | 16.92 a | 92.99 | 1.42 b | 7.80 | 18.20 b |
| | CK | 8.77 a | 96.26 | 0.35 a | 3.81 | 9.11 a | 18.96 a | 85.54 | 3.59 a | 16.20 | 22.16 a |
| Flowering and boll development period | CM1 | 32.61 ab | 80.85 | 7.73 ab | 19.15 | 40.33 ab | 62.63 ab | 71.40 | 25.09 a | 28.60 | 87.72 ab |
| | CM2 | 38.19 ab | 75.78 | 12.07 a | 24.22 | 50.26 a | 63.19 ab | 66.18 | 32.29 a | 33.82 | 95.48 ab |
| | CM3 | 29.36 b | 85.35 | 5.04 b | 14.65 | 34.40 b | 51.39 b | 67.95 | 24.24 a | 32.05 | 75.63 b |
| | CK | 41.48 a | 82.12 | 9.03 ab | 17.88 | 50.51 a | 71.30 a | 72.76 | 26.69 a | 27.24 | 97.99 a |
| Maturation period | CM1 | 49.54 c | 47.81 | 54.09 a | 52.19 | 103.63 b | 78.17 b | 41.65 | 109.54 a | 58.35 | 187.71 ab |
| | CM2 | 64.28 ab | 46.09 | 75.20 a | 53.91 | 139.49 ab | 89.58 b | 49.06 | 92.99 ab | 50.94 | 182.58 ab |
| | CM3 | 52.27 bc | 49.19 | 53.98 a | 50.81 | 106.25 b | 84.91 b | 54.43 | 71.08 b | 45.57 | 155.99 b |
| | CK | 66.78 a | 45.68 | 79.41 a | 54.32 | 146.19 a | 140.44 a | 60.32 | 92.37 ab | 39.68 | 232.81 a |

Note: Different lowercase letters indicate that the difference between different treatments of the same index is significant ($p < 0.05$).

During the same growth period, different allocation patterns affected the distribution ratio of the vegetative and reproductive organs. In 2020, the proportion of dry matter distribution in the reproductive organs of each treatment was as follows: CM2 > CM1 > CK > CM. The distribution and accumulation of dry matter in the CK treatment were the largest at the other growth periods. In 2021, the total weight per plant of the CK treatment was always at the highest level, and the dry matter accumulation at the maturation period was 24.0%, 27.5%, and 33.0% higher than that of CM1, CM2, and CM3, respectively. The dry matter distribution weight of the vegetative organs and reproductive organs in the CM3 treatment was lower than that of the other treatments in each growth period within two years. The reason was that the closer the distance from the jujube tree, the lower the availability of light radiation and the higher the planting density, which increased the shading of the population and limited the growth and dry matter accumulation.

*3.4. Effects of Different Configuration Modes on the Dry Matter Accumulation of the Aboveground and Belowground Parts of the Cotton Plants*

The effects of different configuration treatments on the dry matter accumulation of the aboveground and belowground parts of the cotton plants in the two years are as shown in Figure 10. There are some differences between 2020 and 2021. In 2020, the aboveground dry matter trend was CK > CM2 > CM1 > CM3, and in 2021, it was CK > CM1 > CM2 > CM3. The belowground dry matter trend was consistent with the aboveground dry matter trend. The aboveground dry matter accumulation in each treatment in the two years was as follows: CM1 increased by 43.4 g/plant, resulting in an increase of 42.6%; CM2 increased by 2.7 g/plant, resulting in an increase of 1.9%; CM3 increased by 44.4 g/plant, resulting in an increase of 52.5%; and CK decreased by 5.1 g/plant, resulting in a decrease of 3.0%. By combining and comparing the data from the two years, a synergistic effect between the dry matter accumulation of the aboveground and belowground parts of cotton was observed, whereby greater dry matter accumulation in the belowground parts improved the performance of the aboveground parts. The shaded field environment affected the dry matter accumulation of the belowground and aboveground parts of cotton.

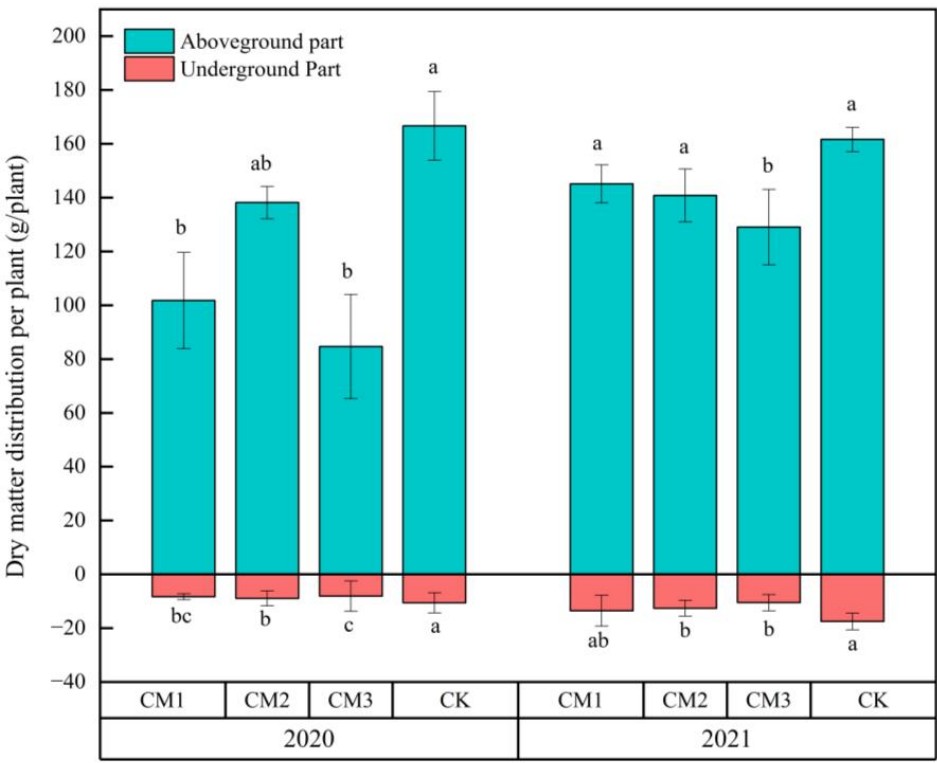

**Figure 10.** Effects of different configuration modes on the dry matter accumulation of the above-ground and belowground parts of the cotton plants. Note: the least significant difference was analyzed in years.

### 3.5. Effects of Different Configuration Modes on Cotton Yield and Yield Components

The two-year cotton yield and yield components were visually displayed on a two-dimensional plane using radar maps (Figure 11). The trends in yield and LER in 2020 and 2021 were consistent. The yield was the highest with CK, in the middle with CM3 and CM2, and the least with the CM1. In 2020, the yield of CK was 145.7%, 67.2%, and 24.2% higher than that of CM1, CM2, and CM3, respectively, and in 2021, it was 163.5%, 69.6%, and 33.8% higher than that of CM1, CM2, and CM3, respectively, indicating that intercropping results in cotton yield reduction. Increasing the number of cotton planting rows in the jujube–cotton intercropping system can increase cotton yield. The LER was the highest with CM2, in the middle with CM3, and the least with CM1 in both years. LER is an indicator used to measure the yield-increasing ability of intercropping patterns relative to monoculture patterns. When LER > 1, it shows that intercropping is more efficient than monoculture. The LER of CM2 and CM3 was greater than 1, indicating that the jujube–cotton intercropping configuration of four rows and six rows was effective, while the configuration of two rows was ineffective. The number of bolls per plant of CM1 was higher in 2020 but was significantly lower than that of the CK treatment in 2021, corresponding to a decrease of 19.41%. There was no significant difference in cotton boll weight and lint percentage among the different configuration modes. The single boll weight was CM2 > CM3 > CK > CM1 in 2020 and CK > CM2 > CM3 > CM1 in 2021, whereas the lint percentage was CK > CM3 > CM2 > CM1 in 2020 and CK > CM1 > CM3 > CM2 in 2021.

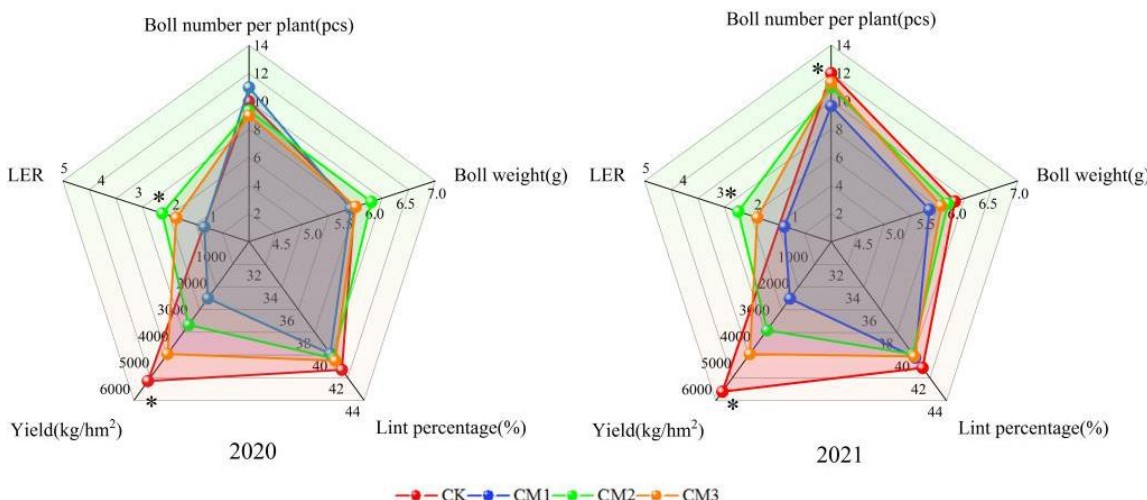

**Figure 11.** Effects of different configuration treatments on cotton yield and yield components. * represents significant difference at the $p > 0.05$ level.

### 3.6. Principal Component Analysis (PCA) of Photosynthetic Characteristic Parameters and Yield and Yield Components

The PCA diagram reflects the relationship between each treatment and each index with PC1 and PC2 (Figure 12). The contribution rate of PC1 was 33.3%, and the contribution rate of PC2 was 23.0%, indicating that the two principal components covered more than 56.6% of the 11 indicators. Rd, LCP, Pmax, LSP, dry weight per plant (Dwpp), boll number per plant (Bpp), lint percentage (Lp), and yield were positively correlated with PC1, indicating that these indicators had a positive contribution to PC1. AQE, LER, and single boll weight (Bw) were negatively correlated with PC1, indicating a negative contribution to PC1. The load coefficient of the Pmax index in PC1 was the largest, and the contribution to PC1 was the largest, followed by LCP, while the Lp contribution was the smallest. The forward projection of Rd in PC2 was the largest, followed by AQE. Similarly, Rd had the greatest positive contribution to PC2 and the highest degree of synergistic change, while AQE ranked second. LER had almost no contribution to PC2 but had a great negative contribution to PC1. The confidence circles of CM3 were crossed with those of the other treatments, indicating that CM3 performed differently from the other treatments in 11 indicators.

A value of 90° between Pmax, Dwpp, LSP, Bpp, yield, and Lp indicates a negative correlation. The 90° angle between Rd and LSP showed that there was no correlation between the two parameters. There was a positive correlation with LCP, AQE, Pmax, and Dwpp, but a negative correlation with yield, Bpp, Lp, LER, and Bw. The PCA diagram also reflected the relationship between each configuration treatment and each index and principal component. The CK treatment was far away from the other treatments, and the difference was large, and it was positively correlated with PC1 and negatively correlated with PC2. This suggests that there was an overall positive impact on PC1 and a negative impact on PC2. There was little difference among the CM1, CM2, and CM3 treatments. CM1 had a positive response to PC1 and PC2, and CM3 had a negative effect on PC1 and PC2. It can be seen that PC1 was greatly affected by the CK treatment, and PC2 was most affected by the CM1 treatment.

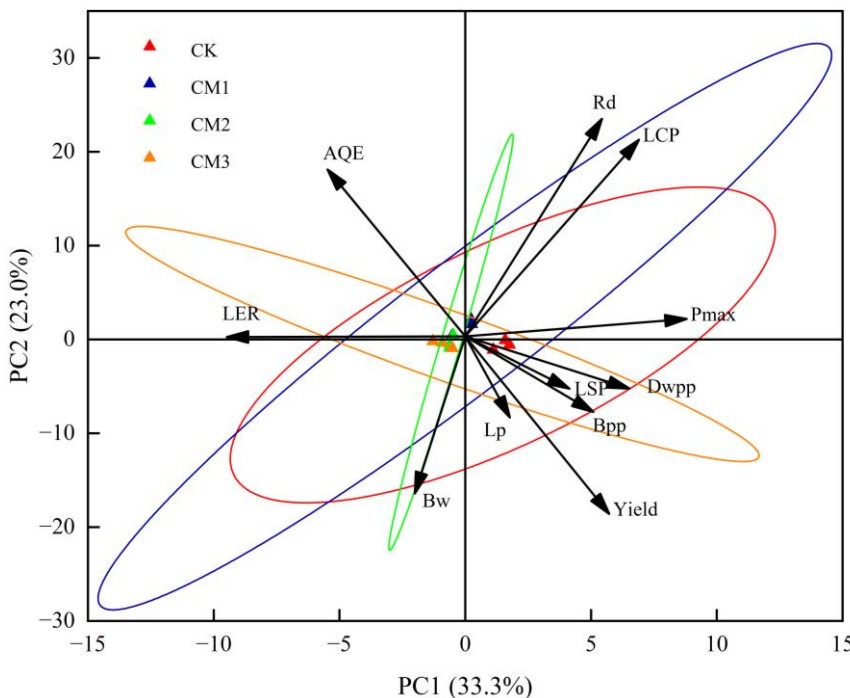

**Figure 12.** Principal component analysis for different configuration treatments. LSP: light saturation point; LCP: light compensation point; Pmax: the maximum net photosynthetic rate; AQE: apparent quantum efficiency; Rd: dark respiration rate; LER: the land equivalent ratio; Bw: single boll weight; Lp: lint percentage; Bpp: boll number per plant; Dwpp: dry weight per plant.

## 4. Discussion

The results showed that with the increase in PAR, the Pn and WUE of each growth period decreased and increased rapidly in the range of low light intensity 0–200 $\mu mol\cdot m^{-2}\cdot s^{-1}$ and even decreased slightly after the inflection point (light saturation point), which indicated that PAR was the main limiting factor of photosynthesis under low light intensity. When the light intensity exceeded a certain value, it began to decrease. With the emergence of strong light, the cotton leaves were unable to absorb and utilize the high light intensity. At the same time, the enzymatic reaction rate in the process of $CO_2$ assimilation limited the photosynthetic rate. The response curve of the CK treatment was the highest and that of CM3 was the lowest in each growth period, which was due to the shade created by the jujube trees. This is consistent with the results of Wang et al. [4]. WUE is an index used to measure the ability of plants to adapt to the environment under the same environmental conditions [34]. The WUE showed a rapid upward trend in the low light intensity range, indicating that the cotton supplemented the photosynthetic reaction substrate by rapidly increasing the WUE and Tr gas exchange rate to adapt to the weak light environment, which was consistent with the Pn results. The light-response curves of Gs and Tr showed a linear relationship, and CM2 obtained better results compared to the other two intercropping treatments. The trend in the Ci response curve was opposite to that of Pn, which decreased rapidly at first and then increased slowly. Huang [35] obtained similar results. The Ci response curve of CM3 was higher than that of the other treatments, indicating that the $CO_2$ concentration of CM3 was high and the utilization rate of light energy was low. Thus, an appropriate intercropping configuration can improve the light-response curve of cotton. A number of studies [6,36] have shown that the closer the distance to the jujube trees, the greater the competition and the smaller and thinner the leaves of edge cotton, as the intercepted light and light-use efficiency of the edge plants are impacted.

The photosynthetic characteristic parameters calculated using the fitting curve can better reflect the utilization efficiency of cotton leaves to light energy. LSP and LCP reflect the adaptability of cotton leaves to strong light and weak light. High LSP and low LCP

mean that cotton leaves have strong adaptability to light intensity, and vice versa [37]. The results showed that the light saturation point LSP of the CK treatment was the highest in the cotton seedling period and budding period, and the LSP of the CM2 treatment was the highest in the flowering and boll development period and maturation period, indicating that the CK and CM2 treatments had strong adaptability to light intensity. With crop growth, the LCP gradually decreased with the CK and CM1 treatments while it decreased first and then increased with the CM2 treatment. Cotton was better able to use strong light at the flowering and boll development period and had the best adaptability to weak light at the seedling period (Table 1). Pmax represents the maximum photosynthetic capacity and the maximum assimilation capacity of leaves [38]. With the aging of the cotton leaves, the Pmax of each treatment was significantly reduced. The CK treatment generated the largest Pmax in each growth period, with the highest value of 33.6. AQE is an important indicator of the slope of Pn-PAR at a low light intensity of 0–200 $\mu mol \cdot m^{-2} \cdot s^{-1}$, and the ability of CM2 to use weak light during the maturation period was significantly higher than that of the other treatments. In the rest of the period, the ability of each treatment to use weak light intensity did not differ much. Rd is related to the physiological activity of the leaves. This study found that Rd and LCP gradually decreased with the decrease in the physiological activity of the leaves, which is similar to the results obtained by Li [19]. It is believed that Rd can increase the utilization and transformation ability of cotton under weak light and reduce the consumption of photosynthetic products to accumulate organic matter, thereby reducing the shading inhibition caused by jujube trees.

The improvement of yield in an intercropping system should not only reduce the light competition in the system but should also promote the reasonable distribution of photosynthetic products to each organ and the accumulation of dry matter. This study found that different configuration modes significantly impacted cotton dry matter accumulation, with the results changing with the growth period. Through two years of investigation, it was found that the dry matter weight distribution of the vegetative organs and reproductive organs in the CM3 treatment were lower than that in the other treatments at different growth periods. In the key growth period of the cotton reproductive organs in 2020, the proportion of dry matter distribution in the reproductive organs of the CM2 treatment was the highest, accounting for 24.2%. In the jujube–cotton intercropping system, the aboveground parts compete for light and heat resources, while the belowground parts compete for water and fertilizer resources. The dry matter accumulation of the aboveground parts and the belowground parts of the cotton plant under different treatments in two years was combined and compared (Figure 10). It was found that the dry matter accumulation of the aboveground parts and the belowground parts had a synergistic effect. The greater the dry matter accumulation in the belowground parts, the better the performance of the aboveground parts. The shading configuration of the six rows affected the accumulation of dry matter in the belowground parts, resulting in long-term concealment between rows and affecting the increase in cotton bolls and the accumulation of dry matter. Xing [39] found that population dry matter increased first and then decreased with increased planting density by equation fitting. The findings of Xing were similar to the present study, and the discrepancy may be because changes in planting structure will directly affect the canopy structure between cotton populations, thereby indirectly affecting the dry matter and yield. We found that the yield of the intercropping cotton reached the best level under a moderate configuration mode, which was consistent with previous research results obtained for monoculture cotton [40,41]. In this study, the yield performance of the two years was CK > CM3 > CM2 > CM1. The yield of the monoculture cotton was significantly higher than that of the intercropping treatments. The yield increased with the increase in intercropping configuration rows. The LER performance was CM2 > CM3 > CM1, reaching up to 2.5 in CM2. Wang et al. [1] found that the two-year average LER (1.564) of a jujube–cotton intercropping system increased productivity and resulted in yield advantages [42] but did not result in a significant difference in lint percentage, single boll weight, and boll number per plant. In the jujube–cotton intercropping system, jujube trees are the dominant

crops, so the shading effect of trees affected the photosynthetic characteristics and yield of intercropped cotton. However, taking intercropping system as a whole, intercropping significantly increased the land-use efficiency, economic benefits, and economic income of farmers because its LER was greater than 1.

The two-row configuration allowed for better overall growth conditions as well as individual advantages. However, due to the lower cotton population in the CM1 treatment, the population photosynthetic rate was lower, the accumulation of photosynthetic substances was smaller, and the yield level was lower compared with the other treatments. The six-row configuration was too large, as the plants were too sheltered by the jujube trees and were also crowding one another due to the row spacing, resulting in limited growth. The photosynthetic rate of the population declined rapidly in the late growth period, and high yields were difficult to obtain in the composite system. The population canopy structure of the four-row configuration mode was reasonable, and sufficient nutrients could be obtained to ensure that more photosynthetic substances were allocated to the reproductive organs. These findings will help farmers to optimize their intercropping system by altering the planting density and the number of rows; for example, by expanding the distance between the cotton boundary row and the jujube trees to minimize the negative impact of the boundary row. The configuration mode proposed in this study can be used to provide a basis for optimizing the spatial configuration in jujube–cotton intercropping systems.

## 5. Conclusions

In summary, the response curves of Pn and WUE of the cotton leaves under different configuration modes at different growth period of a jujube–cotton intercropping system showed an increased trend with the increase in light radiation intensity, with a decrease observed after the inflection point. By contrast, Ci decreased rapidly and then stabilized, whereas Gs and Tr increased linearly. Based on all the factors analyzed, it was concluded that in the jujube–cotton intercropping system, the PAR intercepted by the cotton in the four-row configuration (CM2 treatment) was the highest, the light-response curve was the best, the fitted LSP and Pmax were larger, the utilization rate of strong light was higher, the dry matter distribution and LER were greater compared to the other configuration modes (CM1 and CM3 treatments). Moreover, the canopy structure of CM2 was more reasonable, and the photosynthetic material accumulation was greater than that of the other intercropping treatments. The CM2 mode, which achieves a balance between light resource utilization and economic benefits, provides an effective way to achieve high cotton yields in a jujube–cotton intercropping system in Xinjiang.

**Author Contributions:** Conceptualization, G.C. and S.W.; methodology, T.L., Y.L., and P.W.; software, T.L. and P.W.; validation, G.C.; formal analysis, S.W.; investigation, T.L., and L.L.; resources, G.C. and S.W.; data curation, T.L., R.K., and W.F.; writing—original draft preparation, Y.L., G.C., W.Y., and Y.Z.; writing—review and editing, G.C., Z.F., and Q.W.; visualization, Y.Z.; supervision, S.W.; project administration, G.C.; funding acquisition, G.C. and S.W.; All authors have read and agreed to the published version of the manuscript.

**Funding:** This research was funded by National Natural Science Foundation of China (grant number 32060449,31601272), and the Regional Innovation Guidance Project of Xinjiang Production and Construction Corps (grant number 2021BB012).

**Institutional Review Board Statement:** Not applicable.

**Informed Consent Statement:** Not applicable.

**Data Availability Statement:** The data that support the findings of this study are available from the corresponding authors upon reasonable request.

**Acknowledgments:** The authors sincerely thank the anonymous reviewers who made valuable comments on this paper.

**Conflicts of Interest:** The authors declare no conflict of interest. The funders had no role in the design of the study; in the collection, analyses, or interpretation of data; in the writing of the manuscript, or in the decision to publish the results.

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
