# Peer review of "Effects of Configuration Mode on the Light-Response Characteristics and Dry Matter Accumulation of Cotton under Jujube–Cotton Intercropping"

_applsci, doi:10.3390/app13042427_

Round 1
Reviewer 1 Report
Appreciate the research work and current manuscript but the reviewer has suggestions in the attached files. Hope an improved version.

Reviewer 2 Report
Overall, this is a good paper that needs some minor changes.
Abstract:
Line 16: Please change “away” to “apart”
Introduction:
Line 55: Please change “Ages growth” to “growth”
Line 66: Please remove “also” between “and” and “directly”
Line 100: Please change “curve” to “curves”
Line 103: Please change “provided” to “provide”
Materials and Methods:
Why were there different row spacings for the different treatments? Could row spacing and plant to plant competition contribute to the results significantly?
Line 109: Please change “to” to “with”
Line 136 and throughout the text: Please change “flowering and bolling” to “flowering and boll development” and “bolling” to “maturation”
Line 176: Please change “bell” to “boll”
Line 180: What was the size of the specific area? Did you have different areas based on row spacing?
Results:
Line 206: Please remove the word “basically”
Line 324: Please change the word “at” to “during”
Discussion:
Line 526: Please change “was” to “were”
Line 548: Please change “taken” to “taking”
Figure 8: Please add information about the least significant differences. Are the letters based on 2020 and 2021 or are they analyzed by year?
Conclusions:
Line 572: Please change “increase” to “increased”
